# During Postnatal Ontogenesis, the Development of a Microvascular Bed in an Intestinal Villus Depends on Intussusceptive Angiogenesis

**DOI:** 10.3390/ijms251910322

**Published:** 2024-09-25

**Authors:** Anna V. Zaytseva, Natalia R. Karelina, Eugeny V. Bedyaev, Pavel S. Vavilov, Irina S. Sesorova, Alexander A. Mironov

**Affiliations:** 1Department of Anatomy, Saint Petersburg State Pediatric Medical University, Saint Petersburg 194100, Russia; karelina_nr@mail.ru; 2Department of Anatomy, Ivanovo State Medical University, Ivanovo 153012, Russia; akb37@mail.ru (E.V.B.); vavilov-007@mail.ru (P.S.V.); irina-s3@yandex.ru (I.S.S.); 3Department of Cell Biology, IFOM ETS—The AIRC Institute of Molecular Oncology, Via Adamello, 16, 20139 Milan, Italy

**Keywords:** angiogenesis, intestine, intestinal villus, intussusceptive angiogenesis, postnatal development

## Abstract

The mechanisms responsible for the growth and development of vascular beds in intestinal villi during postnatal ontogenesis remain enigmatic. For instance, according to the current consensus, in the sprouting type of angiogenesis, there is no blood flow in the rising capillary sprout. However, it is known that biomechanical forces resulting from blood flow play a key role in these processes. Here, we present evidence for the existence of the intussusception type of angiogenesis during the postnatal development of micro-vessel patterns in the intestinal villi of rats. This process is based on the high-level flattening of blood capillaries on the flat surfaces of intestinal villi, contacts among the opposite apical plasma membrane of endothelial cells in the area of inter-endothelial contacts, or the formation of bridges composed of blood leucocytes or local microthrombi. We identified factors that, in our opinion, ensure the splitting of the capillary lumen and the formation of two parallel vessels. These phenomena are in agreement with previously described features of intussusception angiogenesis.

## 1. Introduction

The intestinal villus is a unique structure, spread out in the anteroposterior direction and covered with suction enterocytes, which are distinguished by the presence of a special shape of the APM with a brush border formed by microvilli. Along with these enterocytes, the epithelial lining contains goblet-shaped cells that secrete mucus. The function of the intestinal villus is determined not only by the special structure of the intestinal epithelium but also by the peculiar structure of the vascular network. In adult mammals, the arteriole that feeds the villus cells goes from the arteries of the submucosal zone directly to the tip of the villi, where it divides into two micro-vessels, i.e., pre-capillaries running along both crests of the villi on the front and back surfaces of the villi [1]. These pre-capillaries give off numerous capillary anastomoses, which form a fairly dense network. The vessels of the network merge into veins formed near the base of the villi on the anterior and posterior surfaces. This is a general description, but there may be variations in the structure. There are one or two–four lymphatic capillaries in the center of the villi. Such an organization of blood circulation in the villi of the small intestine implements a counter-current mechanism of movement and oxygen exchange between the central ascending arteriole and descending parallel venules. This ensures the maintenance of a relatively low partial pressure of oxygen at the tip of the villi. The action of the counter-current mechanism also leads to the formation of an osmotic gradient in the intestinal villi with the highest osmolarity (salt concentration) at the tip. However, the counter-current mechanism is not implemented in all species. In contrast, in newborns, the microvascular network of the intestinal villi is often represented by a single vascular loop [1].

How an abundant microvascular structure forms from a loop is practically unknown. Previously, Karelina [2] believed that this is due to sprouting. Recently, we analyzed in detail the mechanisms of angiogenesis [3]. Among them, the main ones are sprouting, growth by forming a vascular hairpin, and microvascular cleavage (links from angiogenesis). In this article, we tried to determine which type of angiogenesis described in our previous article is involved in the construction of a vascular structure that feeds the intestinal villi.

Blood capillaries surround the lymphatic capillaries and directly attach to the basement membrane (BM) of enterocytes. The highly fenestrated part of the capillary wall faces this BM [1]. However, blood capillaries cannot absorb chylomicrons because of the fact that fenestrae are blocked by a protein membrane, and the size of the holes in the membrane is significantly smaller than the diameter of the chylomicrons. Absorption of chylomicrons is realized by lymphatic capillaries [4,5,6,7,8,9,10,11,12,13,14].

The formation of new vessels occurs from existing ones. Currently, two main mechanisms of angiogenesis have been described as follows: intussusceptive angiogenesis and the formation of growth buds or sprouts and the splitting of capillaries into two. Also, the so-called “ansiform” angiogenesis single-angioblastic processes that branch off the apex of capillary hairpins or loops are described. The sprouting mechanism is well known. It starts when the local concentration of a ligand or several ligands for EC receptors located on the BLPM EC of the maternal vessel is increased. This local site could be induced by the action of tumor cells or chemicals (for example, crystal of AgNO_3_) or another type of tissue damage, where angiogenesis will then begin. Then, the diffusion of a ligand or several ligands towards the micro-vessel begins from this focus (usually a postcapillary venule or capillary lined with continuous endothelium with caveolae, which could provide additional space for growing sprouts or protrusion of the plasma membrane (PM) [15,16,17]). However, we did not find sprouts in the microvascular bed of intestinal villi newborn rats [1]. 

Intussusceptive angiogenesis is described by Burri’s group [18,19]. It is the process by which pre-existing vessels split or remodel through the formation of transluminal tissue pillars, leading to expansion (intussusceptive microvascular growth), arborization (intussusceptive arborization), or branching remodeling (intussusceptive branching remodeling) of the pre-existing vasculature. This type of angiogenesis has been described in different animal models, during both normal and pathological microvascular growth, including the development of the lung, kidney, ovary, retina, bone, and skeletal muscle, among other tissues and organs, in the rapidly expanding pulmonary capillary bed of neonatal rats, in different organs, and even in tumors. Intussusceptive angiogenesis permits rapid expansion of the capillary plexus, when there is a limitation in capillary growth by sprouting [20,21,22,23]. Intussusceptive angiogenesis was described during pathology in the colon but not in the intestinal villi [20,21,22]. However, these papers are not related to the small intestine or its intestinal villi. On the other hand, there are a lot of papers related to pathology but not to normal development. 

In spite of this, there are no systematic studies documenting the postnatal development of the microvascular bed in the intestinal villi [4,11,12,13,14]. In this paper, we tried to study changes in the vascular structures of the intestinal villi during postnatal ontogenesis and find signs of either sprouting or splitting of capillaries.

## 2. Results

In newborns, depending on the structure of the inflection site of the loop-shaped micro-vessel, the following vascular structures can be distinguished: (1) a loop with a hairpin-shaped end, (2) a loop with a hairpin-shaped end and the presence of a median anastomosis more or less parallel to the axis of the villi, (3) a hairpin-shaped loop with an anastomosis connecting the adductor and abductor knees and located perpendicular to the axis of the villi, and (4) an arcuate loop with formed marginal micro-vessels running along the ridges and the presence of pronounced anastomosis between them. 

After birth, some casts of their proximal part were often split into two. The two micro-vessels formed in this area can anastomose with each other and even merge again. On serial semi-thin sections of the villi of the jejunum of newborn animals, it is clearly visible that a vascular loop is located at the very tip of the villi directly under the epithelium. In the first section, it has the appearance of one very large oval-shaped micro-vessel. As you move towards the basal part of the villi, the loop section of the lumen becomes longer and then divides into two independent micro-vessels. There are signs of an ongoing angiogenetic process. The graph is the number of holes per unit length of the micro-vessel projection. The structure of the microvascular bed in intestinal villi of newborn rats is shown in Figure 1A–D,G1. The red arrows show the holes in the DAB-positive projection of micro-vessels and casts.

Using stereology, we calculated the number of such “holes” per unit length of micro-vessels in the total samples labeled with diaminobenzidine (DAB). Corrosion castes cannot be used for these purposes since they do not show all micro-vessels but only their surface on one side. We created a graph of the proportion of flattened capillaries on semi-thin sections, showing the % of sections with sharply compressed flattened capillaries on the proximal and distal surfaces of the forming villi. The graph in Figure 2 demonstrates that the fraction of the villi containing signs of capillary splitting increased up to the 10th day. Their patterns were similar for each sign of vessel splitting. After feeding, especially on the second day of life, the middle part of the villi increases dramatically in size and the villi acquires a barrel shape [1].

By 3 days after birth, serial semi-thin sections of the villi of the jejunum of newborn animals clearly show that a vascular loop is located at the very tip of the villi directly under the epithelium. In the first section, it has the appearance of one very large oval-shaped micro-vessel. As you move towards the basal part of the villi, the loop section lengthens and then divides into two independent micro-vessels. At this level, other micro-vessels appear, usually of a smaller diameter. Sections of blood micro-vessels surrounding the lymphatic capillary are divided into two populations. The larger ones go along the crest of the villi. Moreover, the section of one of them (the venous knee) has a 1.5 times larger diameter than the other. Micro-vessels of smaller diameter are located under the epithelium on the cranial and caudal surfaces of the flattened intestinal villi. Differentiation of marginal (going along the ridges) blood micro-vessels and subepithelial capillaries becomes more pronounced. The adducting and diverting knees differ significantly in diameter. At the base of the villi, the centrally located lymphatic is surrounded by four micro-vessels. Most of the villi consist of the following two parts: a conical, flattened base and a cylindrical, less flattened apical part. The vessels on the casts become sharply convoluted. The anastomoses between the knees of the vascular loop move closer to the tip of the vascular loop. On semi-thin sections, blood capillaries on the distal and proximal surfaces of a villus are represented by a narrow slit-like vessel at the very base of the intestinal villi. With the help of serial semi-thin sections, it was found that in one-day-old rats, the blood capillaries form a loop, which is divided into venular and arteriolar knees and preserved.

The structure of microvascular beds in the intestinal villi of the three-day rats is shown in Figure 1F–H,M,N. The low flattened part of the intestinal villi is demonstrated in Figure 1N. The number of holes in the DAB-positive projections of the microvascular bed is high. The red arrows show the holes in the DAB-positive projections of micro-vessels and casts. The semithin sections of the jejunal villi at the middle level are shown in Figure 1O,R,S. The red arrows indicate the section of flattened capillaries. In the semithin sections parallel to the longitudinal axis of a villus, the flattened capillaries could be detected, although with difficulty (Figure 1T).

Figure 3 demonstrates the intussusception type of angiogenesis in rat intestinal villi in the 3-day rats. The serial block-face SEM (3VIEW) images (arrows) show flattened capillaries. Figure 4 presents examples of the serial images, which demonstrate flattened capillaries (arrows) with the island in the middle of the proximal jejune villus. Figure 4 shows the transmission EM analysis of flattened capillaries on the flat surfaces of intestinal villi. Figure 5A shows a highly flattened capillary (red arrow). Figure 5B shows a microthrombus (arrow) in the flattened capillary. Figure 5C shows a blood cell in the flattened capillary. Figure 5D indicates contact and fusion among lamellipodia in the flattened capillary (arrow). Figure 5E presents the opposite contact zones in the capillary able to form splits. Figure 5F,G show the interaction of lamellipodia from opposite inter-endothelial contacts.

On the 10th day of life, leaf-shaped villi appear among the conical villi. At the same time, in both types of villi, it is already possible to distinguish a centrally running arteriole. In the first case, it flows into one of the knees of the marginal micro-vessel, leaving its apical part free for anastomosis with capillaries; in the second case, it flows into the horizontal apical part of the marginal micro-vessel eccentrically. Microvascular structures of these types of villi differ in that in the first case, their structure approaches that of conical villi. The casts have a wide base formed by an elongated perpendicular plexus and a densely packed intensely anastomosing conglomerate of vessels at the tip of the villi. The last cluster of vessels is somewhat narrowed at the point of transition to a wide vascular base and resembles vascular glomeruli. 

The structure of microvascular beds in intestinal villi of the 10-day rats is shown in Figure 1M,O,R. The number of holes in the DAB-positive projections of microvascular beds and casts remained high. The red arrows show the holes in the DAB-positive projection of micro-vessels and casts.

By the end of the first month of life, the formation of the differentiated microvascular plexus of the intestinal villi is almost complete. In monthly animals, the structure of the villi almost completely coincides with that of adults. The vast majority of villi have a trapezoidal shape. However, some of them have a longer upper edge, and the other part is shorter, which brings them closer in shape to the lingual villi. The structure of microvascular structures in both types of villi is almost identical and similar to that in adults. On the cranial and caudal surfaces of the villi, the subepithelial vascular network has a pronounced loopy character; the loops are usually hexagonal in shape. The vast majority of villi have a trapezoidal shape. However, some of them have a longer upper edge, and the other part is shorter, which brings them closer in shape to the lingual villi. The subepithelial networks on the cranial and caudal surfaces of the flattened villi have a regular hexagonal character, but the collective venules are weakly expressed and very short. The terminal micro-vessels running along the crests of the villi have a sharply convoluted shape. The microvascular structure of the villi of the jejunum in an adult rat corresponds to our previous descriptions. On the cranial and caudal surfaces of the villi, the subepithelial vascular network has a pronounced loopy character; the loops are usually hexagonal in shape. Almost the same design of the microcirculatory bed is revealed in the villi of mature animals. At the same time, the area of fenestrae fields increases more and more. Dense connections become quite well-developed. Thirty days after birth, the differentiation process of the microcirculatory bed is complete. In the endothelium of capillaries, the formation of basement membranes, fenestra fields, and dense connections with inter-endothelial contacts ends [14]. Up to 30 days after birth, the general trend in their development is that the villi grow in length and flatten. Mostly flattened capillaries are observed on the distal and proximal surfaces of a villus.

The structure of microvascular beds in intestinal villi of the 30-day rats is shown in Figure 1P,S. The number of holes in the DAB-positive projections of the microvascular beds decreased. The red arrows show the holes in the DAB-positive projection of these micro-vessels and casts.

## 3. Discussion 

In this work, we investigated the development of blood micro-vessels supplying the intestinal villi in the process of postnatal ontogenesis and showed that the process does not go through clearly defined stages. Rather, the stages are more blurred, and at each stage, there are several forms of microvascular networks at once. 

Our data indicated that during the formation of a very characteristic vascular bed in intestinal villi, intussusceptive angiogenesis could be the main mechanism. Here, we presented several types of evidence suggesting that the maturation of microvascular beds in the intestinal villi occurs based on capillary splitting. Indeed, we found round pores in the plastic casts obtained during postnatal ontogenesis. These pores were mostly observed on the cranial and caudal surfaces of the cats of intestinal villi. Also, in the 3VIEW (serial block-face scanning electron microscopy) samples, these holes were visible. After perfusion fixation, the capillaries on these surfaces were often compressed and contained blood cells. In the images obtained with the help of serial block-face scanning electron microscopy, we observed islands on the lumen of these capillaries. In the newborn rats, areas with high-density fenestra were not observed. 

If immediately after birth the villi have a cylindrical shape and are supplied with blood through one loop-shaped vessel, then by the third day after birth, intestinal villi with a cylindrical tip and a flattened base appear. By day 10, cylindrical villi with a single vascular loop disappear and flattened leaf-shaped villi predominate; however, villi in the diene of a flattened base with a cylindrical top also persist. Thirty days after birth, almost all intestinal villi are leaf-shaped. The micro-vessel pattern is already very similar to the vascular bed in adult animals. A study of micro-vessel designs labeled with DAB or filled with Mercox resin showed that immediately after birth, there are rounded “holes” in the vessels that are not filled with blood containing horseradish peroxidase, which oxidizes DAB, visible as an osmophilic agent, and Mercox resin. Their number increases up to 10 days, and then, by the 30th day, the relative density of such “holes” begins to fall. No such formations were found in adults. These are signs of angiogenesis based on capillary cleavage. We studied semi-thin sections, images obtained during the sampling of serial images of the cut block inside the block on a scanning electron microscope, and randomly selected images of capillary sections adjacent to the flat surfaces of intestinal villi. 

The dynamics of the appearance (numerical density) of such images during the process of postnatal ontogenesis corresponded to the dynamics that we found in the process of studying the projections of micro-vessels and castes filled with Mercox under SEM labeled with DAB and osmium. Flattened capillaries were often found between two opposite apical PMs. This usually occurred in the area of inter-endothelial contacts, which were characterized by the presence of lamellipodia and filopodia protruding into the lumen. Several molecules and blood flow participate in the regulation of intussusceptive angiogenesis. Variable expression of VEGF and VEGF receptors (VEGFR1 and VEGFR2) may condition sprouting or intussusceptive angiogenesis. Hypoxia-inducible factor 2 alpha, angiopoietins 1 and 2, FGF (fibroblast growth factor 2), and platelet-derived growth factor beta (PDGF beta) also act in intussusceptive angiogenesis regulation. Blood flow changes play an important role in vascular plexus remodeling and differentiation of venules and arterioles [23,24,25,26].

The hypothesis of intussusceptive angiogenesis poses that pre-existing vessels split or are remodeled through the formation of transluminal tissue pillars, leading to the expansion (intussusceptive microvascular growth), arborization (intussusceptive arborization), or branching remodeling (intussusceptive branching remodeling) of the pre-existing vasculature. This type of angiogenesis has been described in different animal models during both normal and pathological microvascular growth; during the development of the lung, kidney, ovary, retina, bone, and skeletal muscle, among other tissues and organs, in the rapidly expanding pulmonary capillary bed of neonatal rats; in different organs; and even in tumors. This process starts when contact between the APMs of ECs from opposing capillary walls is formed. Then, inter-endothelial cell junctions are subjected to reorganization, and an interstitial pillar core is formed that is invaded by pericytes. By this stage, transluminal pillars have a diameter of about 2.5 µm. Then, a hole through this contact is formed. Pillar growth leads to a rapid expansion of the capillary plexus. Intussusceptive microvascular growth permits the rapid expansion of the capillary plexus, furnishing a large endothelial surface for metabolic exchange. Intussusceptive arborization causes changes in the size, position, and form of preferentially perfused capillary segments, creating a hierarchical tree [27,28,29,30,31,32,33,34,35,36,37,38,39,40,41].

When a blood capillary is confined to a single EC, the EC is referred to as a seamless EC. Seamless ECs make up about 50% of all capillaries in the renal glomeruli. Seamless ECs are found in arterio-venous capillaries in the endocrine glands, as well as in the sinusoidal systems of the heart muscle, liver, spleen, and bone marrow. As glomerular maturation proceeds, the capillary loop becomes divided into six to eight loops. Most of the capillaries in the glomeruli of the kidney are seamless [41,42]. Terasaki et al. [43] demonstrated that during glomerulus maturation, both intussusceptive angiogenesis and sprouting occur. This suggests that during the differentiation of these glomeruli, the main mechanism of angiogenesis is splitting. 

Several molecules and blood flow participate in the regulation of intussusceptive angiogenesis. The variable expression of VEGF and VEGF receptors, namely, VEGFR1 and VEGFR2, may regulate sprouting or intussusceptive angiogenesis. Hypoxia-inducible factor 2 alpha, angiopoietin 1 and 2, FGF (fibroblast growth factor 2), and platelet-derived growth factor beta (PDGF beta) also act in intussusceptive angiogenesis regulation. Blood flow changes play an important role in vascular plexus remodeling and the differentiation of venules and arterioles [44,45,46,47,48,49,50,51,52,53,54,55,56,57,58,59,60,61,62,63,64].

Based on our observations and taking into consideration the literature data, we proposed the following model for the development of capillary-splitting angiogenesis. Figure 6 demonstrates the possible mechanism of intussusception angiogenesis in the intestinal villus. In Figure 6A–G, we show schemes of bed reorganization. In Figure 6H, we show a scheme of capillary splitting.
There should be a restriction in the space for the growth of micro-vessels, which would lead them to flatten, as in the glomeruli of the kidney, where there is a sharp restriction in the volume of the glomerulus by the surrounding convoluted tubules or in the alveoli of the lungs, where only vascular growth is possible in one plane. The limitation of the intestinal villi is their flat surfaces, the appearance of which is due to the fact that the intestine contracts longitudinally.The flattening of capillaries only leads to the gluing of their APMs when the contact zones of the EC are opposed to each other, which contributes to the gluing of lamellipodia and cylindrical outgrowths of the APMs.The contact of the outgrowths of the ECs leads to their fusion and the formation of an empty space penetrating the capillary. Then, inter-endothelial cells are subjected to reorganization. A hole is formed through these contacts. Continuous pillar formation and growth lead to a rapid expansion of the capillary plexus [29,30]. The presence of blood cells attached to this zone or local deposition of fibrin contributes to the cleavage of the capillary lumen.Perhaps this attracts connective tissue cells, which, having penetrated there, fix the cleavage of the capillary lumen.

Thus, we believe that intussusceptive angiogenesis starts when contact between the APM of ECs from opposing capillary walls is formed. 

## 4. Material and Methods

The methodology and details of all ethics rules were described in [1,2]. Briefly, all experimental animal procedures were approved by the Committees of the Ivanovo State Medical Academy. Wistar rats were obtained from the Moscow Cardiological Center (they obtained them from Taconic Farms (Germantown, NY, USA)) and were maintained either on Purina rodent chow (no. 5001 ICN Pharmaceuticals, Inc., Cleveland, OH, USA) or using manually prepared food corresponding to the standards. All procedures were in accordance with EU directive 2010/63/EU. The animal facilities of the St. Petersburg Pediatric University and Ivanovo State Medical University housed the animals in plastic sawdust-covered cages on a 12 h dark/light cycle, keeping them under standard conditions (at room temperature and fed standard rat pelleted food and water ad libitum). 

Six Wistar newborn male rats were taken immediately after birth, and six Wistar newborn rats were examined when their stomachs appeared full after feeding. Also, the rats were taken for experiments 3, 10, and 30 days after birth. Each time, we used 6 male rats. The rats were anesthetized with a combination of Zoletil (containing the active substances zolazepam hydrochloride and thiamine hydrochloride in equal proportions) and 2% Rometar (the active ingredient is xylazine hydrochloride) in a ratio of 3:1 at a dose of 0.1 mL per 100 g of body weight. The animals were removed from the experiment before the end of anesthesia after opening the chest by the intracranial administration of a saturated solution of potassium chloride at a dose of 1–2 mM/kg [1,2].

Trained persons sacrificed the rats as described [1]. Briefly, death was confirmed by observing cessation of heartbeat and respiration and absence of reflexes, in agreement with international standards (https://www.lal.org.uk, access date: 20 March 2022). While the animals were under ether anesthesia, jejune tissue was removed, processed, embedded, sectioned, and stained. All experimental animal procedures were approved by the Committees of the Ivanovo State Medical Academy and St. Petersburg State Pediatric University. The procedures for animal use were conducted in accordance with the ethical and legal standards of the Russian Federation mentioned in Order no. 755 of the Ministry of Health of the USSR of 12 August 1977, “On measures to further improve the organizational forms of work using experimental animals”, and a letter from the Ministry of Agriculture dated 5 February 2022 no. 13-03-2/358, “On modern alternatives to the use of animals in the educational process”, and 2010/63/EU legislation on animal protection. The experiments were approved by the decision of the Academic Council of St. Petersburg Pediatric University no. 10 from 23 September 2015 and the decision of the ethics committee of Ivanovo State Medical Academy (no. 1 from 5/XII, 2018) in compliance with the above-mentioned Order no. 755 of the Ministry of Health of the USSR of 12 August 1977, “On measures to further improve the organizational forms of work using experimental animals”, and a letter from the Ministry of Agriculture dated 5 February 2022 no. 13-03-2/358, “On modern alternatives to the use of animals in the educational process”. All experiments on live animals were carried out in Russia; samples were irreversibly fixed with glutaraldehyde and embedded in Epon or gelatin (with subsequent fixation) in Russia, and only then were the plastic samples transported to Italy, where they were examined. 

Trained persons sacrificed rats. Death was confirmed observing cessation of heartbeat and respiration, and absence of reflexes, in agreement with international standards (https://www.lal.org.uk; access date: the 20 March 2022). While the animals were under ether anesthesia, jejune tissue was removed, processed, embedded, sectioned and stained. After injection of anesthetic, the abdomen of the animal was opened and the initial part of the jejune was cut and placed into fixative. All those experimental animal procedures were approved by the Committees of the Ivanovo State Medical Academy (see ethic statements in [1,2]).

Semithin and ultrathin sections were prepared and examined as described previously [65,66]. Briefly, after the fixation of the samples with 2.5% glutaraldehyde in 0.1 M cacodylate buffer (pH 7.4), they were post-fixed. Initially, the samples were washed with 0.15 M sodium cacodylate buffer followed by incubation in reduced OsO_4_ for 1 h on ice. After washing in distilled water, the samples were incubated again in 0.3% thiocarbohydrazide for 20 min, washed with distilled water, and finally incubated a third time in 2% OsO_4_ in water for 30 min.

The microvascular corrosion casting/SEM method and horseradish peroxidase-based labeling of blood vessels were performed as described [1,14]. Briefly, the animals were anesthetized as described above. The casting media was injected into the heart using a syringe. The injection pressure was equal to 80 mm Hg. Complete removal of the blood was performed. The circulatory system was rinsed with 37 °C heparinized Tyrode^®^ solution (5000 IU/L) until the efflux of the incised portions of several subcutaneous venae of animal limbs was clear. Immediately after the pre-casting treatment, the injection medium was prepared by adding to the main reagent (resin), a catalyst/accelerator, to initiate polymerization. Thereafter, we injected Mercox II Red (from LARD Research Industries; catalog number 21,245–Mercox II Red–Kit). The animal bodies were left for two hours at room temperature and then put in a 60 °C water bath for 24 h to complete the polymerization of the perfused casting medium. The small intestines were removed and macerated in 15% KOH solution at 40 °C for 2 days. The specimens were cleaned in 2% formic acid. In order to remove the surrounding tissues, the injected specimens were immersed into the 15–20% sodium hydroxide or potassium hydroxide solution (at 60 °C, for 24 h). In order to remove the white saponified materials resulting from the maceration of tissues rich in lipids with sodium hydroxide, the casts were washed in running water. Then, microdissection was carried out to expose the structures of interest. The vascular casts were dried in the air without any detectable distortion or dislocation of the fragile parts of the casts. In some cases, we used freeze-drying to minimize surface tension. The specimens were coated with a heavy metal (gold) and then observed in SEM with an accelerating voltage of 5–10 kV. Then, SEM observation of the corrosion casts was performed. 

Six quadruplets of animals composed of 1 control and 3 experimental rats were formed. In two random samples taken from the control and all experimental rats, we calculated the percentage of villi containing signs of intussusceptive angiogenesis. Then, the average percentage of positive samples was calculated for the control adult rats and for the newly born, 3-day, 10-day, and 30-day rats. These mean values were considered the statistical units. Thus, for each sign of intussusceptive angiogenesis, we had four statistical units representing the average percentage for each time point. In order to estimate the percentage of cells with a defined phenotype, we used blind analysis and examined 3–4 cells in each member of the pair. To test whether differences were significant (*p* < 0.05), Student’s *t*-tests, paired *t*-tests, and non-parametric Mann–Whitney U tests were used. The normality of the datasets was assessed using Shapiro–Wilk normality tests. In the majority of cases, we used nonparametric Mann–Whitney U tests. Values are mean ± SD of 6 variants (*n* = 6). A difference was considered significant when *p* < 0.05. The standard software package GraphPad Prism (Prism: Version 9.4.2) was used. Data are given as means ± standard deviations (SDs). In the text, the words “differ”, “smaller”, and “higher” indicate that two values are significantly (*p* < 0.05) different [2,67].

## Figures and Tables

**Figure 1 ijms-25-10322-f001:**
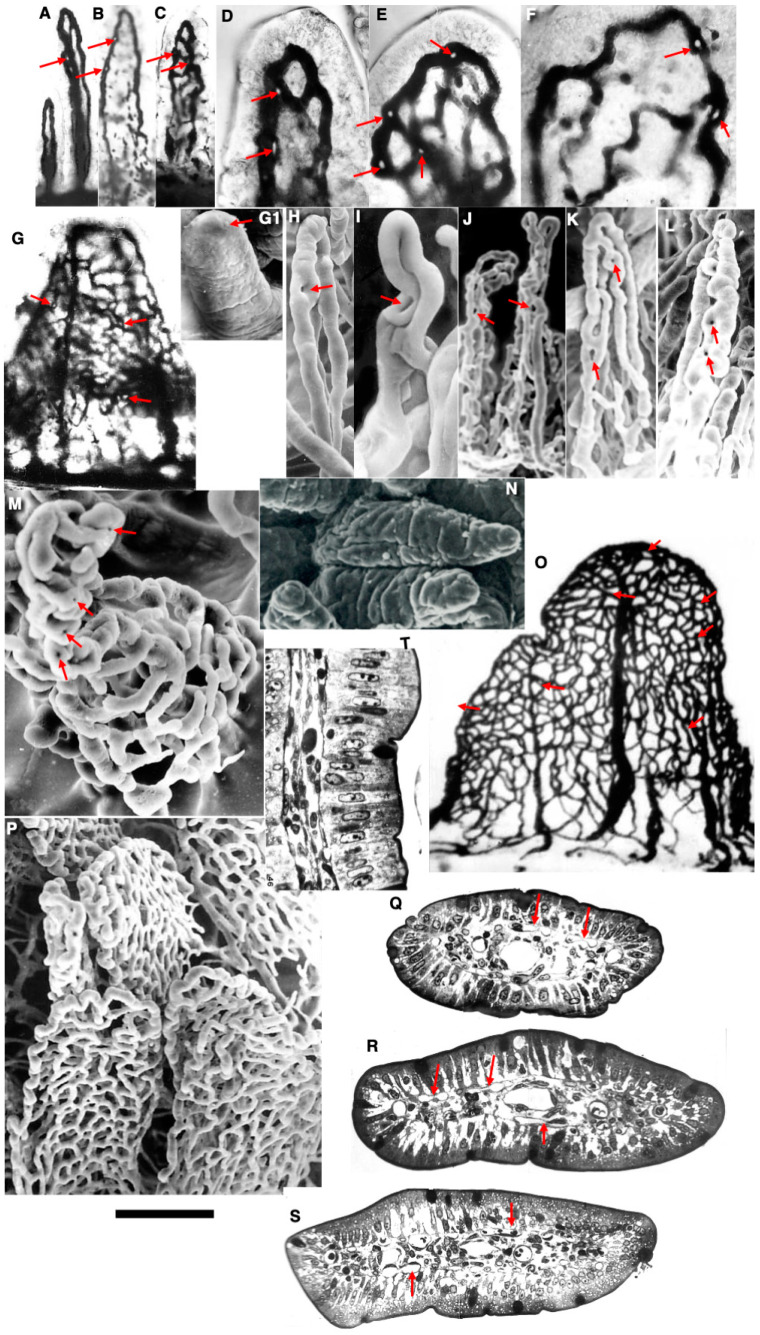
Structure of microvascular beds in the intestinal villi of rats. (**A**–**D**,**G1**) Newborn rats. (**E**–**H**,**M**,**N**) Three-day rats. (**M**,**O**,**R**) Ten-day rats. (**P**,**S**) Thirty-day rats. O, Red arrows show the holes in the DAB-positive projection of micro-vessels and casts. (**N**) The low part of the intestinal villi became flattened. (**O**) By 10 days, the number of holes in the DAB-positive projections of the microvascular bed was high. (**P**) By 30 days, the number of holes in the casts decreased. (**G**–**T**) Semithin section of jejunal villi at the middle level. The red arrows indicate the section of flattened capillaries. (**T**) In the semithin section parallel to the longitudinal axis of a villus, the flattened capillaries could be detected rarely. (**A**) The images are taken from Figure 5G presented by Nikonova et al. [1]. (**C**) The image is taken from Figure 5H presented by Nikonova et al. [1]. (**H**–**K**) The images are taken from Figure 5B–E presented by Nikonova et al. [1]. Figures reprinted courtesy of a Creative Commons License (Attribution–Noncommercial–Share Alike 4.0 Unported license). The bars is 10 µm.

**Figure 2 ijms-25-10322-f002:**
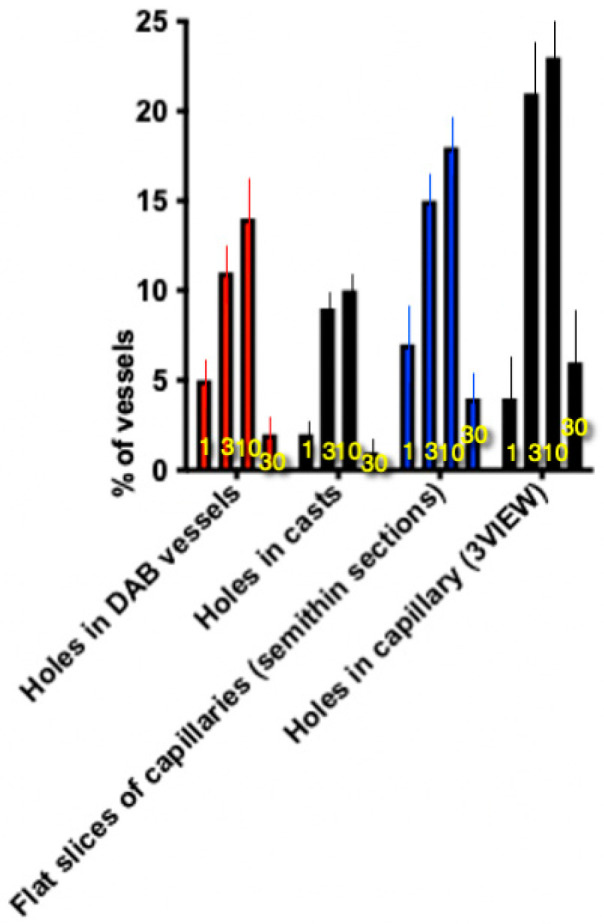
Percentage of different signs of capillary splitting. Numbers indicate days after the birth. Means ± SD are shown. In all cases bar 1 is significantly (*p* < 0.05) smaller than bar 10, whereas bar 10 is larger than bar 30.

**Figure 3 ijms-25-10322-f003:**
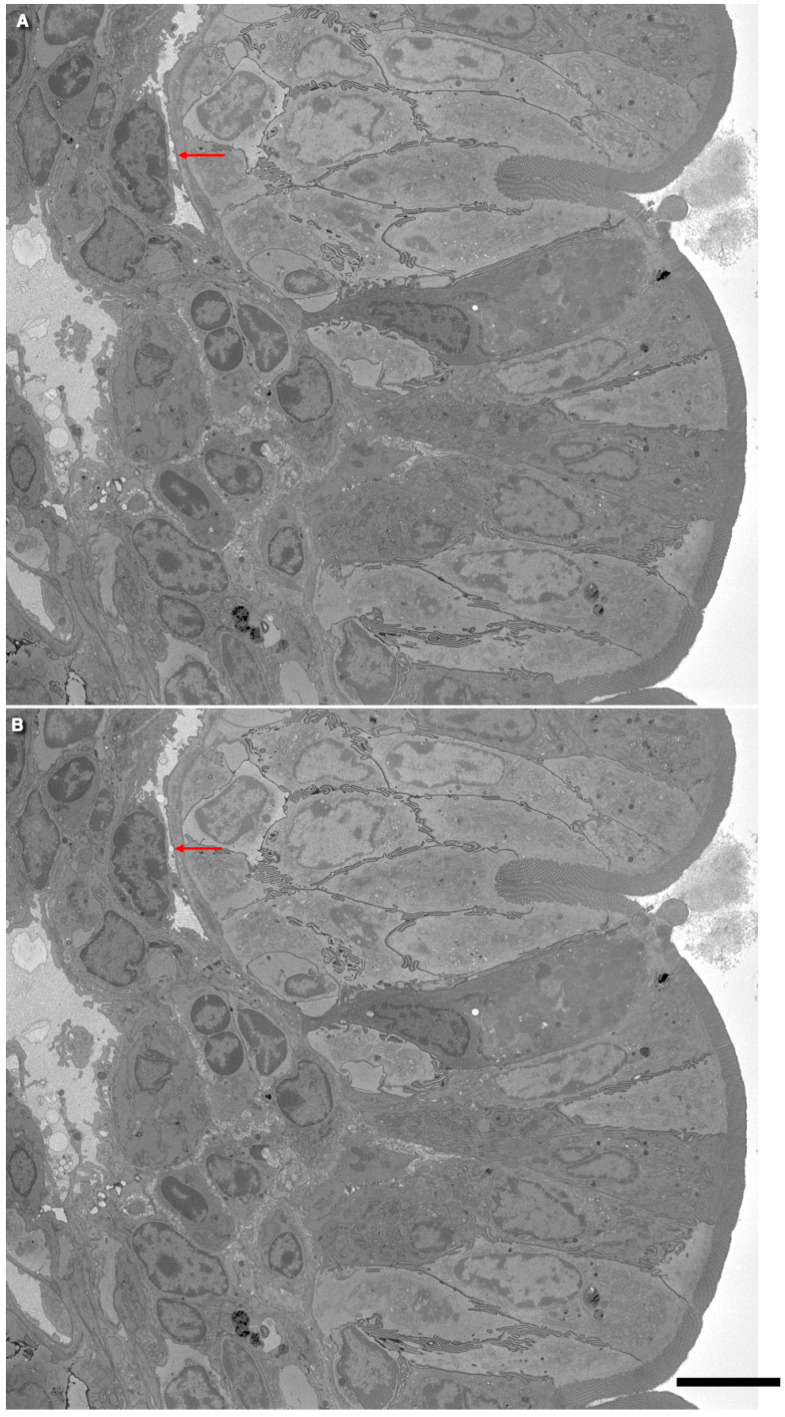
Serial images of a flattened capillary (arrows). In (**A**,**B**), red arrows indicate the places where the capillary lumen would be merged. The intussusception type of angiogenesis in rat intestinal villi in 3-day rats. Serial block-face SEM (3VIEW). The bars is 5 µm.

**Figure 4 ijms-25-10322-f004:**
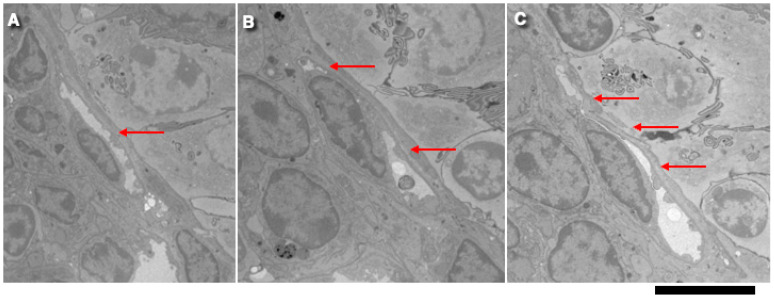
Intestinal villi of the three-day rats. Serial images demonstrate flattened capillaries (arrows show lumen) with an island in the middle. Images (**A**–**C**) represent serial sections of the same capillary. In (**C**), under high magnification, one can see that it is one vessel. The bar is 5 µm.

**Figure 5 ijms-25-10322-f005:**
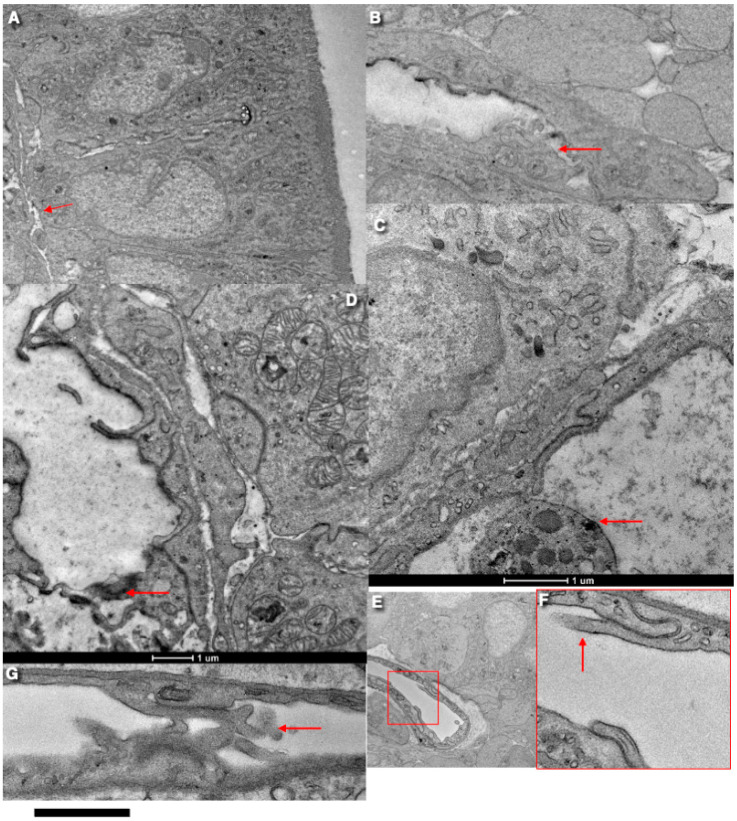
Analysis of flattened capillaries on the flat surfaces of intestinal villi. Different mechanisms of intussusceptive angiogenesis. (**A**) Highly flattened capillary (red arrow). (**B**) A microthrombus (red arrow) in the flattened capillary. (**C**) A blood cell in the flattened capillary. (**D**) Contact and fusion between lamellipodia in the flattened capillary (arrow shows a local microthrombus). (**E**) Opposite contact zones in the capillary able to form splits. (**F**) Interaction of lamellipodia (red arrow) from opposite inter-endothelial contacts. Numbers indicate how many days passed after birth. (**G**) Contacts of endothelial cell protrusions (red arrows) from opposite sides of the capillary. Bars: 500 nm.

**Figure 6 ijms-25-10322-f006:**
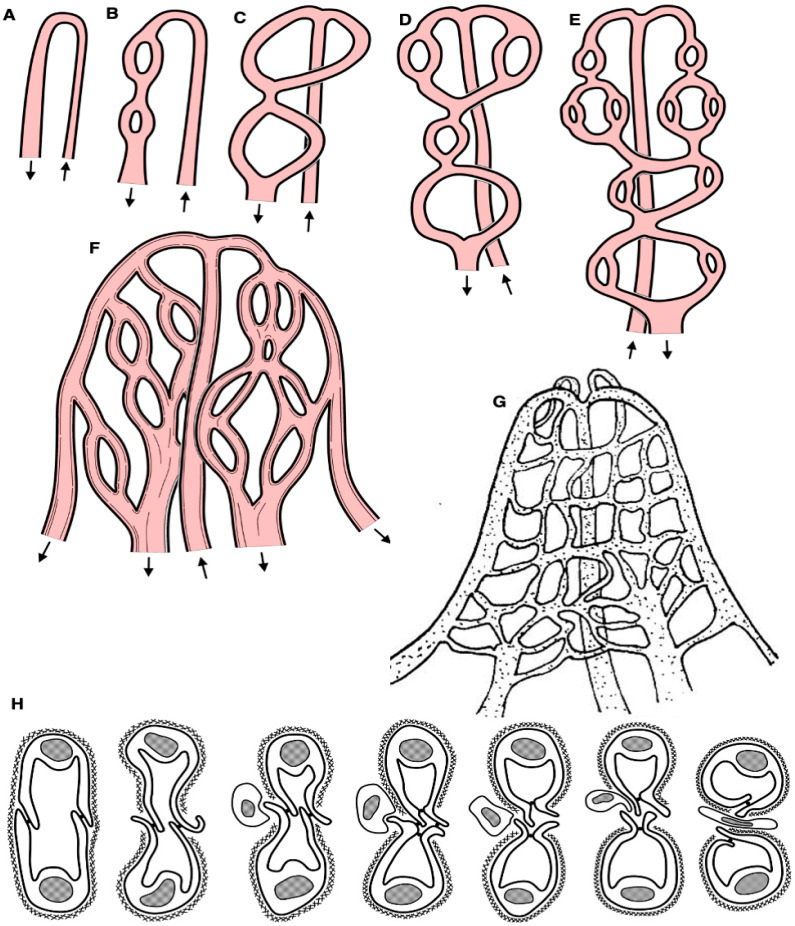
Possible mechanism of intussusception angiogenesis in the intestinal villus. (**A**–**G**) Schemes of bed reorganization. (**H**) Scheme of capillary splitting. Arrows indicate the direction of the blood flow.

## Data Availability

Data is contained within the article.

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
