# Peer review of "During Postnatal Ontogenesis, the Development of a Microvascular Bed in an Intestinal Villus Depends on Intussusceptive Angiogenesis"

_ijms, 2024, doi:10.3390/ijms251910322_

Round 1

Reviewer 1 Report

Comments and Suggestions for Authors
  • Major points:
  • The authors morphologically identified "holes" in the blood vessels of rat neonatal intestinal villi tissue. By comparing different stages, they argue that intussusceptive angiogenesis occurs in neonatal intestinal villi.
  • Intussusceptive angiogenesis in the intestine has been reported in many papers, particularly in pathological conditions such as colitis, and is widely recognized. However, the authors did not address these previously documented phenomena.
  • The authors claim to have identified elements of capillary lumen splitting and the formation of two parallel vessels during intussusceptive angiogenesis. However, the supporting experimental data for these claims were not presented.
  • The manuscript completely lacks a Materials and Methods section. The authors refer to their previous paper, stating, "All reagents, procedures, and analyses were performed s previously described [1]." However, this manuscript fails to provide any information on the analytical methods employed, statistical analysis, sample sizes, or any other details that would allow for assessment of the experimental data generation methods or the significance of the analytical results.
  • While describing histological structural differences, no quantitative data was presented. It is unclear whether they are referring to a single characteristic section or if the statements are based on statistical analysis of a larger dataset from many sections. This issue also came from the absence of the Materials and Methods section.
  •  
  • Minor points:
  • The order of the figure descriptions in the Results section is out of the order in Figure layouts, making them difficult to read, would recommend that reconsidering the structure and order of the figure data.
  • The text "The graph" at line 187 on page 4 did not specify any Figure nor data.
  • The meaning of "DAB-positive projection" are not described through the manuscript. This is due to the lack of the Materials and Methods section. Please describe all the experimental methods performed in this manuscript.
  • There is no explanation of what B-G are in the legend of Figure S1.
  • There is no explanation for A and B in Figure 2.
  • There is no explanation of the differences between A-F in Figure 3.
  • In Figure 4G, it is not described how many rats were used for the analysis, how many photographs and how many measurement points were used, and how the categories of histological differences were distinguished. There is no description of how statistical significance was calculated.

Author Response

Reply

We indicated our changes with blue colour. We increased the number of references in order to decrease the percentage of ours.

Reviewer 1

The authors morphologically identified "holes" in the blood vessels of rat neonatal intestinal villi tissue. By comparing different stages, they argue that intussusceptive angiogenesis occurs in neonatal intestinal villi.

Major points:

  1. Intussusceptive angiogenesis in the intestine has been reported in many papers, particularly in pathological conditions such as colitis, and is widely recognized. However, the authors did not address these previously documented phenomena.

Our reply

These papers are not related to small intestine and its intestinal villi. On the other hand, there are a lot of the papers related to pathology and not to normal development. Thus, formally speaking these papers are not related to our topic and therefore were not quoted. However, we added these references.

Esteban S, Clemente C, Koziol A, Gonzalo P, Rius C, Martínez F, Linares PM, Chaparro M, Urzainqui A, Andrés V, Seiki M, Gisbert JP, Arroyo AG. Endothelial MT1-MMP targeting limits intussusceptive angiogenesis and colitis via TSP1/nitric oxide axis. EMBO Mol Med. 2020 Feb 7;12(2):e10862. doi: 10.15252/emmm.201910862.

However, in this paper did not prove that this type of angiogenesis is really present. They just hypothesized this. Even in their casts typical holes are not visible.

Konerding MA, Turhan A, Ravnic DJ, Lin M, Fuchs C, Secomb TW, Tsuda A, Mentzer SJ. Inflammation-induced intussusceptive angiogenesis in murine colitis. Anat Rec (Hoboken). 2010 May;293(5):849-57. doi: 10.1002/ar.21110.

Intussusceptive angiogenesis was described during pathology in colon and not in intestinal villi.

Ackermann M, Tsuda A, Secomb TW, Mentzer SJ, Konerding MA. Intussusceptive remodeling of vascular branch angles in chemically-induced murine colitis. Microvasc Res. 2013, 87, 75-82. doi: 10.1016/j.mvr.2013.02.002.

Intussusceptive angiogenesis was described during pathology and not in intestinal villi

  1. The authors claim to have identified elements of capillary lumen splitting and the formation of two parallel vessels during intussusceptive angiogenesis. However, the supporting experimental data for these claims were not presented.

Our reply

We presented serial black face SEM images obtained using 3VIEW (see Fig. 3). Red arrows show the same capillary in serial images when initially it appeared as a single vessel, then, it is divided into two vesseals, and finally, it became a single vessel.

  1. The manuscript completely lacks a Materials and Methods section. The authors refer to their previous paper, stating, "All reagents, procedures, and analyses were performed s previously described [1]." However, this manuscript fails to provide any information on the analytical methods employed, statistical analysis, sample sizes, or any other details that would allow for assessment of the experimental data generation methods or the significance of the analytical results.

Our reply

It was a mistake. In the revised version of the paper which we sent after demand of our editor we added this chapter.

  1. While describing histological structural differences, no quantitative data was presented. It is unclear whether they are referring to a single characteristic section or if the statements are based on statistical analysis of a larger dataset from many sections. This issue also came from the absence of the Materials and Methods section.

Our reply

In the previously revised version these data were present (see explanation above).

Minor points:

  1. The order of the figure descriptions in the Results section is out of the order in Figure layouts, making them difficult to read, would recommend that reconsidering the structure and order of the figure data.

Our reply

We corrected this.

  1. The text "The graph" at line 187 on page 4 did not specify any Figure nor data.

The meaning of "DAB-positive projection" are not described through the manuscript. This is due to the lack of the Materials and Methods section. Please describe all the experimental methods performed in this manuscript.

There is no explanation of what B-G are in the legend of Figure S1.

There is no explanation for A and B in Figure 2.

There is no explanation of the differences between A-F in Figure 3.

In Figure 4G, it is not described how many rats were used for the analysis, how many photographs and how many measurement points were used, and how the categories of histological differences were distinguished. There is no description of how statistical significance was calculated.

Our reply

We corrected these mistakes.

Reviewer 2 Report

Comments and Suggestions for Authors

In "During postnatal ontogenesis the development of microvascular bed in an intestinal villus depends on the intussusceptive angiogenesis", Zaitseva and colleagues describe anatomical features of the development of the vascular bed in the intestinal villus of neonate rats. The authors use a histological approach to obtain morphological evidence that is used to generate a model of the developmental process. The present work fills a gap in studies of the intestinal villus that have been done largely in adult animals. Below are some comments and suggestions for the authors:

  1. The introduction covers well the current knowledge of the structure of the vascular beds in the intestine of different animal species. However, the authors could improve this section by reducing its length. For example, detailed anatomical descriptions are needed, but references to previous work and reviews are too elaborate and could be more concise. I would suggest keeping the first two paragraphs and rewriting the rest into a third paragraph that explains the need for the present study.
  2. In the opinion of this reviewer, a supplementary figure is not needed for the introduction. Reference to previous studies should be sufficient for the specialized readers. If the authors are concerned about the general readership, they can consider making a figure that summarizes the anatomical features of the vascular bed that are relevant for the present study. For example, one panel could be a micrograph showing a representative structure, and another panel could be a cartoon indicating the features of interest for the present study.
  3. The results section can be improved. For example the first five paragraphs in the results section seem a continuation of the introduction. I would advise to rewrite them into one paragraph that can be used to introduce the results of figure 1. Note that some of the points made in the present version of the paper can also be addressed in the discussion.
  4. I would suggest to the authors to organize the results section into subheadings that make specific points. Currently it is difficult to follow the logic of result presentation, even though the micrographs in the figures are of great quality and with relevant information.
  5. Please check that arrows and other figure markings point to the right structures. For example, the leftmost arrow in figure 1 O seem to be offset.
  6. The materials and methods section is well written and contains information that will be useful for those wanting to reproduce the results of this study.
  7. The discussion section is concise and in agreement with the observations. I would advise against increasing the length of the discussion.
Comments on the Quality of English Language

Editorial corrections (punctuation and style) are needed throughout all sections of the manuscript.

Author Response

Reviewer 2

In "During postnatal ontogenesis the development of microvascular bed in an intestinal villus depends on the intussusceptive angiogenesis", Zaitseva and colleagues describe anatomical features of the development of the vascular bed in the intestinal villus of neonate rats. The authors use a histological approach to obtain morphological evidence that is used to generate a model of the developmental process. The present work fills a gap in studies of the intestinal villus that have been done largely in adult animals. Below are some comments and suggestions for the authors:

  1. The introduction covers well the current knowledge of the structure of the vascular beds in the intestine of different animal species. However, the authors could improve this section by reducing its length. For example, detailed anatomical descriptions are needed, but references to previous work and reviews are too elaborate and could be more concise. I would suggest keeping the first two paragraphs and rewriting the rest into a third paragraph that explains the need for the present study.

Our reply

We corrected the text and reduced the length of the introduction

  1. In the opinion of this reviewer, a supplementary figure is not needed for the introduction. Reference to previous studies should be sufficient for the specialized readers. If the authors are concerned about the general readership, they can consider making a figure that summarizes the anatomical features of the vascular bed that are relevant for the present study. For example, one panel could be a micrograph showing a representative structure, and another panel could be a cartoon indicating the features of interest for the present study.

Our reply

We eliminated supplementary Figures.

  1. The results section can be improved. For example, the first five paragraphs in the results section seem a continuation of the introduction. I would advise to rewrite them into one paragraph that can be used to introduce the results of figure 1. Note that some of the points made in the present version of the paper can also be addressed in the discussion.

Our reply

We changed our text according to this recommendation.

  1. I would suggest to the authors to organize the results section into subheadings that make specific points. Currently it is difficult to follow the logic of result presentation, even though the micrographs in the figures are of great quality and with relevant information.

Our reply

We corrected our text.

  1. Please check that arrows and other figure markings point to the right structures. For example, the leftmost arrow in figure 1 O seem to be offset.

Our reply

We checked images.

  1. The materials and methods section is well written and contains information that will be useful for those wanting to reproduce the results of this study.

Our reply

We added some new information about counting

  1. The discussion section is concise and in agreement with the observations. I would advise against increasing the length of the discussion.

Our reply

We increased its length

Comments on the Quality of English Language

Editorial corrections (punctuation and style) are needed throughout all sections of the manuscript.